# *Ziziphus nummularia*: A Comprehensive Review of Its Phytochemical Constituents and Pharmacological Properties

**DOI:** 10.3390/molecules27134240

**Published:** 2022-06-30

**Authors:** Joelle Mesmar, Rola Abdallah, Adnan Badran, Marc Maresca, Abdullah Shaito, Elias Baydoun

**Affiliations:** 1Department of Biology, American University of Beirut, Beirut P.O. Box 11-0236, Lebanon; jm104@aub.edu.lb (J.M.); rha62@mail.aub.edu (R.A.); 2Department of Nutrition, University of Petra, Amman 961343, Jordan; abadran@uop.edu.jo; 3Aix-Marseille University, CNRS, Centrale Marseille, iSm2, 13013 Marseille, France; 4Biomedical Research Center, College of Medicine and Department of Biomedical Sciences at College of Health Sciences, Qatar University, Doha P.O. Box 2713, Qatar

**Keywords:** *Ziziphus nummularia*, phytochemicals, cyclopeptide alkaloids, nummularine-M, antioxidant, anti-inflammatory

## Abstract

*Ziziphus nummularia*, a small bush of the Rhamnaceae family, has been widely used in traditional folk medicine, is rich in bioactive molecules, and has many reported pharmacological and therapeutic properties. **Objective**: To gather the current knowledge related to the medicinal characteristics of *Z. nummularia*. Specifically, its phytochemical contents and pharmacological activities in the treatment of various diseases such as cancer, diabetes, and cardiovascular diseases, are discussed. **Methods**: Major scientific literature databases, including PubMed, Scopus, ScienceDirect, SciFinder, Chemical Abstracts, Medicinal and Aromatic Plants Abstracts, Henriette’s Herbal Homepage, Dr. Duke’s Phytochemical and Ethnobotanical Databases, were searched to retrieve articles related to the review subject. General web searches using Google and Google scholar were also utilized. The search period covered articles published between 1980 and the end of October 2021.The search used the keywords ‘*Ziziphus nummularia*’, AND (‘phytochemical content’, ‘pharmacological properties, or activities, or effects, or roles’, ‘anti-inflammatory’, ‘anti-drought’, ‘anti-thermal’, ‘anthelmintic’, ‘antidiabetic’,’ anticancer’, ‘anticholinesterase’, ‘antimicrobial’, ‘sedative’, ‘antipyretic’, ‘analgesic’, or ‘gastrointestinal’). **Results**: This plant is rich in characteristic alkaloids, especially cyclopeptide alkaloids such as nummularine-M. Other phytochemicals, including flavonoids, saponins, glycosides, tannins, and phenolic compounds, are also present. These phytochemicals are responsible for the reported pharmacological properties of *Z. nummularia*, including anti-inflammatory, antioxidant, antimicrobial, anthelmintic, antidiabetic, anticancer, analgesic, and gastrointestinal activities. In addition, *Z. nummularia* has anti-drought and anti-thermal characteristics. **Conclusion**: Research into the phytochemical and pharmacological properties of *Z. nummularia* has demonstrated that this plant is a rich source of novel bioactive compounds. So far, *Z. nummularia* has shown a varied pharmacological profile (antioxidant, anticancer, anti-inflammatory, and cardioprotective), warranting further research to uncover the therapeutic potential of the bioactives of this plant. Taken together, *Z. nummularia* may represent a new potential target for the discovery of new drug leads.

## 1. Introduction

Plants have been widely used in healthcare to treat many diseases since prehistorical times. The increase in population size, lack of some medications and often high cost, as well as the unwanted side effects observed with many synthetic drugs, has led to increased scientific and commercial attention to medicinal plants. In fact, many effective drugs used nowadays are derived from plant sources. For example, morphine, which is used to relieve severe pain, is derived from *Papaver somniferum* plant or the opium poppy [1]. Other examples include digoxin, which is used in treating heart failure, is derived from *Digitalis purpurea* or foxglove plant [2] and quinine from *Cinchona* bark, which is used in treating malaria [3]. With the huge medicinal success of herbs, the development of plant-based drugs will continue.

The genus *Ziziphus* of the Rhamnaceae family contains over 58 accepted species of thorny shrubs and small trees growing mostly in arid and semi-arid regions. Species of this genus are traditionally known for their health benefits, nutritional values, and therapeutic properties, as described by various cultures around the world, specifically in India, Pakistan, China and the Middle East [4,5,6]. Although research on the health benefits of *Ziziphus* is still evolving, its potential uses range from reducing inflammation, improving gastrointestinal health, controlling diabetes, preventing or treating neurological diseases, treating dry skin and sunburns, promoting wound healing, reducing wrinkles, and having an anti-infective and anti-cancer potential. The *Ziziphus* species are in fact a rich source of bioactive compounds. Analysis of their phytochemical composition revealed about 431 chemical constituents so far, including, alkaloids (particularly cyclopeptide alkaloids), flavonoids, terpenoids, saponins, and other minor compounds such as cholinergic acids, aromatic or polyaromatic compounds, steroids, cerebrosides and nucleosides [4]. 

*Ziziphus jujuba*, *Ziziphus mauritiana*, *Ziziphus nummularia*, *Ziziphus spina-christi*, *Ziziphus lotus*, and *Ziziphus xylopyrus* are the most studied species of the genus. The literature describes, more or less extensively, their ethnobotanical, phytochemical and pharmacological attributes. Of particular interest to this review is *Ziziphus nummularia*, which is a small thorny branched bush native to India, Pakistan, and Iran, growing mainly in arid and dry areas (Figure 1). It is widely used in traditional folk medicine. Its fruit is used as food; it has a cooling effect and removes biliousness [7], and its ripened fruit powder has been reported to treat constipation [8]. Its leaves are used in the treatment of scabies and other diseases of the skin [5], conjunctivitis [9], diarrhea [9], gastrointestinal spams [10], and helminthiasis [5]. The roots are used for the treatment of dysentery [11] and the stem bark is used for alleviating joint pain as well as sore throat and bleeding gums [12,13]. There have also been reports on its beneficial role in treating insomnia, anxiety and fever, although these have not yet been supported by scientific proof [14]. 

The therapeutic value of *Ziziphus nummularia* was reported by several other reviews, where various biological properties were reported such as its antioxidant, anti-inflammatory, antimicrobial, antiproliferative, hypoglycemic and hypolipidemic activities [4,15,16,17]. Moreover, numerous phytochemicals have been isolated from various parts of the plant, including tannins, flavonoids, steroids, glycosides, and alkaloids [18]. This richness in bioactive molecules justifies its use in the prevention and/or treatment of a wide spectrum of pathological conditions as described above, and calls for further investigation into the impending benefits of *Ziziphus nummularia*. This review aims to provide a comprehensive overview of the phytochemical properties of *Ziziphus nummularia* and its pharmacological activities based on an up-to-date examination of the literature, shedding light on the cellular and molecular mechanisms involved. All told, the findings exposed here will advance the potential of *Ziziphus nummularia* as an attractive herbal agent for future drug discovery.

## 2. Scientific Classification (Index Kewensis)

Kingdom: Plantae; Phylum: Tracheophyta; Class: Magnoliopsida (Dicotyledons); Order: Rosales; Family: Rhamnaceae; Genus: Ziziphus Mill; Species: *Ziziphus nummularia*; Binomial name: *Z. nummularia* (Burm.fil.) Wight & Arn. 

## 3. Methods

Major scientific literature databases, including PubMed, Scopus, ScienceDirect, SciFinder, Chemical Abstracts, Medicinal and Aromatic Plants Abstracts, Henriette′s Herbal Homepage, Dr. Duke′s Phytochemical and Ethnobotanical Databases, were searched to retrieve articles related to the review subject. General web searches using Google and Google scholar were also utilized. The search period covered articles published between 1980 and the end of October 2021. The search used the keywords and MeSH terms for ‘*Ziziphus nummularia*’, AND (‘phytochemical content’, ‘pharmacological properties, or activities, or effects, or roles’, ‘anti-inflammatory’, ‘anti-drought’, ‘anti-thermal’, ‘anthelmintic’, ‘antidiabetic’,’ anticancer’, ‘anticholinesterase’, ‘antimicrobial’, ‘sedative’, ‘antipyretic’, ‘analgesic’, or ‘gastrointestinal’). 

## 4. Phytoconstituents of *Zizyphus nummularia*

*Zizyphus nummularia* is rich in several phytochemical compounds, particularly cyclopeptide alkaloids. One of the earliest studies performed in 1983 reported the isolation of two new cyclopeptide alkaloids, nummularine-M and nummularine-N (Table 1 and Table 2), and the presence of a dozen cyclopeptide alkaloids in the root bark of the plant [19]. Further research by the same team led to the identification of other cyclopeptide components: nummularogenin in 1984 [20], nummularine-P in 1987, and nummularine-S in 1988 in the stem bark (Table 1 and Table 2) [21,22]. Cyclopeptide alkaloids are natural macrocyclic compounds with fascinating chemical and biological characteristics, widely present in the Rhamnaceae family, and particularly the *Ziziphus* genus [23,24]. In fact, these compounds have been described as playing a role in the central nervous system due to their sedative, analgesic, and anti-nociceptive effects, in addition to having antimicrobial properties and an antidiabetic potential [23], This has been drawing the attention of chemists and biologist to elucidate the structure-activity relationships of these compounds, being a promising source of therapeutic agents.

The use of different solvents in the process of extraction and plant part can lead to the identification of different phytochemicals. For instance, in a study investigating the composition of *Ziziphus nummularia* leaves, the authors showed that 105 constituents were identified in the n-hexane extract consisting mainly of terpenoids, and 56 compounds were isolated from the ethanolic extract, mostly alkaloids and flavonoids with ethyl alpha-d-glucopyranoside as the main constituent (Table 1 and Table 2) [25]. In another study using *Ziziphus nummularia* fruits, the dichloromethane and petroleum ether extracts showed the highest flavonoid content, compared to methanolic and acetone extractions, with the latter exhibiting a high phenolic content [26]. Moreover, a comparison of the phytochemical composition between the leaves and fruits of various *Ziziphus nummularia* populations revealed that leaves were richer in phenols saponins and flavonoids [27]. And when it comes to the essential oils of this plant, these were shown to be rich in monoterpenes, aliphatic hydrocarbons, alkane hydrocarbons, primary terpene compounds, and decarbonated alcohols (Table 1 and Table 2) [28]. The major constituents of *Ziziphus nummularia* plant parts using various extraction solvents are summarized in Table 1, which emphasizes the importance of this plant as a rich source of bioactive ingredients.

**Table 1 molecules-27-04240-t001:** Summary of the phytochemical composition of extracts from different parts of the *Zizyphus nummularia,* using different solvents.

Plant Part	Solvent Used	Main Results	Major Compounds	References
**Root bark**	Benzene	Isolation of new peptide alkaloids containing the 14-membered ring system	Amphibin-H	[29,30,31,32]
			Nummularine-A	
			Nummularine-B	
			Nummularine-C	
			Nummularine-D	
			Nummularine-E	
			Nummularine-F	
			Nummularine-G	
			Nummularine-K	
			Nummularine-H	
			Mucronin-D	
	Methanol	Identification of new cyclopeptide alkaloids and peptide alkaloids	Jubanine-A	[30]
			Jubanine-B	
			Mauritine-C	
**Stem bark**	Methanol	Identification of new cyclopeptide alkaloids and peptide alkaloids	Amphibinine-H	[19,21,22,30,33,34,35,36]
			Frangufoline	
			Fubanine-B	
			Jubanine-B	
			Nummularine-B	
			Nummularine-M	
			Nummularine-N	
			Nummularine-O	
			Nummularine-P	
			Nummularine-R	
			Nummularine-S	
			Nummularine-T	
			Nummularine-U	
			Mauritine-A	
			Mauritine-F	
			Mauritine-D	
			Scutianine-C	
**Leaves**	Petroleum ether	Detection of glycosides, and saponins		[37]
	Ethanol	Isolation of a pure saponin	Zizynummin	[38]
	Ethanol	Detection of alkaloids, glycosides, and saponins		[37]
		Presence of carbohydrates, protein, alkaloids, phenol, flavonoids, tannins, and saponins		[39]
		Identification of 56 phytoconstituents. Ethyl alpha-d-glucopyranoside polysaccharide was the main constituent. Some phytosterols and fatty acids were also extracted.	2 methoxy-4-vinylphenol	[25]
			Ethyl alpha-d- glucopyranoside Behenyl behenate	
			Linoleic acid	
			Gamma sitosterol	
			Stigmasterol	
			Phytol	
			Squalene	
			Pleic acid	
			Tricosane	
			Tetradecane	
	n-hexane	Presence of carbohydrates, protein, alkaloids, phenol, flavonoids, tannins, saponins, and glycoside, in addition to fixed oils, fats, and volatile oils		[39]
		Identification of 105 phytoconstituents, mainly terpenoids. A few fatty acids and some phytosterols have also been extracted	1-eicosanol	[25]
			Betulin	
			Campesterol	
			Gamma sitosterol	
			Geranyl linalool isomer	
			Linoleic acid	
			Lupeol	
			Palmitic acid	
			Phytol	
			Stearic acid	
			Squalene	
			Stigmasterol	
			Vitamin E	
			Trans-geranylgeran oil	
	Hydro-alcoholic	Detection of saponins, flavonoids, glycosides, tannins, and phenolic compound. Protein and amino acid were absent	Quercetin	[40]
	Methanol	Presence of saponins, triterpenes and flavonoids Sequential fractionation of the crude methanolic extract was carried out with n-hexane, chloroform, ethyl acetate, and water.		[41]
**Roots**	Methanol	Presence of alkaloids, tannins, terpenoids, reducing sugars, saponins, flavonoids, steroids, cardiac glycosides, coumarins, emodines, and anthocyanins and betacyanins in the crude extract. Anthraquinones, glycosides, and phlobatanins were absent. Solvent fractionation of the crude methanolic extract was carried out with n-hexane, chloroform, ethyl acetate, and ethanol		[42]
**Fruits**	Hydro-alcoholic	Detection of flavonoids, glycosides, tannins, and phenolic compound Proteins and amino acid were absent	Quercetin	[40]
	Acetone	High phenolic content and presence of flavonoids and flavonols		[26]
	Dichloromethane	Very high flavonoid content and presence of phenolics		[26]
	Hydro-alcoholic	Detection of flavonoids, glycosides, tannins, and phenolic compound Proteins and amino acid were absent	Quercetin	[40]
	Methanol	Presence of phenolics and flavonoids in equal amounts		[26]
		Identification of different phenolic compounds from different plant genotypes	Chlorogenic acid	[43]
			Hydroxy benzoic acid	
			Quercetin	
			Mandelic acid	
			Morin	
			Pyrogallol	
			Rutin	
**Whole plant**	Benzene	Isolation of a new (25 S)-spirostane: nummularogenin	Nummularogenin	[20]
	Methanol	Identification of 45 compounds with high-resolution mass spectra in positive and negative ionization modes	Guanidinosuccinic acid	[44]
			Daidzin	
			N-isovaleroylglycine	
			Guaiphenesin	
			Sucrose	
			Quinic acid	
			Coumaroylquinic acid	
			Myricetin-3-O-galactoside	
	Essential oils	39 chemical compounds were isolated form the plant, mainly: monoterpenes, aliphatic hydrocarbons, alkane hydrocarbons, primary terpene compounds, and decarbonated alcohol	Tetradecane	[28]
			Hexadecane	
			dl-limonene	
			Cyclohexan-1-ol	
			3 meth	
			Trans-caryophyllene	
			Beta-myrcene	

**Table 2 molecules-27-04240-t002:** Chemical structures of the main phytoconstituents in the stem bark, root bark, leaves, and fruits of *Ziziphus nummularia*.

Class	Name	Structure	Description	References
**Alkaloids**	Nummularine D	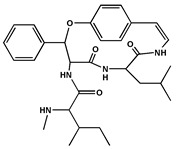	Cyclopeptide alkaloid isolated from the root bark	[29]
Nummularine N	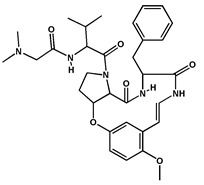	Cyclopeptide alkaloid isolated from the stem bark	[19]
Nummularine P	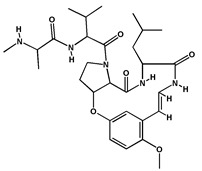	Cyclopeptide alkaloid isolated from the stem bark	[21]
Nummularine R	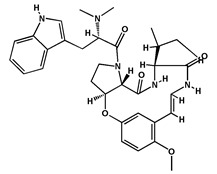	Peptide alkaloid isolated from the stem bark	[34]
Jubanine B	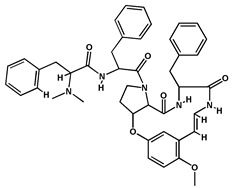	Peptide alkaloid isolated from the root bark and stem bark	[30,33]
**Terpenoids**	Geranyl linalool	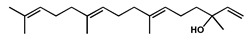	Diterpenoid isolated from the leaves	[25]
Squalene	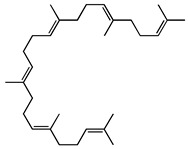	Triterpene isolated from the leaves	[25]
Lupeol	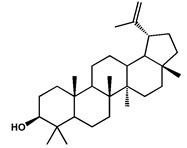	Triterpene isolated from the leaves	[25]
Phytol	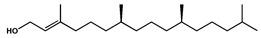	Diterpene alcohol isolated from the leaves	[25]
IC	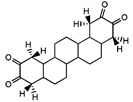	A triterpene derivative (octadecahydro-picene-2,3-14-15-tetranone) isolated from the root bark with in vitro activity against human breast cancer, leukaemia, ovarian cancer, colon adenocarcinoma, and human kidney carcinoma. It also showed in vivo anticancer activity in mice againt Ehrlich ascites carcinoma	[45]
**Flavonoids**	Chlorogenic acid	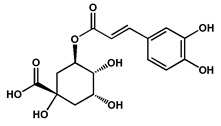	Polyphenol isolated from the fruits, suggested to play an important role in the management of Alzheimer’s disease by potentially inhibiting acetylcholinesterase	[43]
Quercetin	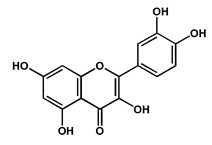	Polyphenol isolated from the fruits and leaves, suggested to play an important role in the management of Alzheimer’s disease by potentially inhibiting acetylcholinesterase	[40,43]
Morin	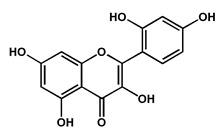	Phenolic compound isolated from the fruits in high concentrations, suggested to play an important role in the management of Alzheimer’s disease by potentially inhibiting acetylcholinesterase	[43]
Pyrogallol	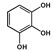	Polyphenol isolated from the fruits	[43]
Rutin	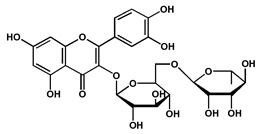	Glycoside isolated from the fruits, suggested to play an important role in the management of Alzheimer’s disease by potentially inhibiting acetylcholinesterase	[43]
**Glycosides**	Ethyl alpha-d-glucopyranoside	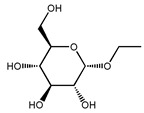	Glycoside isolated from the leaves	[25]
**Benzenoids**	Mandelic acid	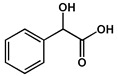	Alpha hydroxy acid isolated from the fruits	[43]
**Quinones**	Lapachol	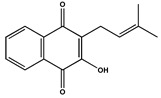	Naphtoquinone isolated from the plant and was shown to have strong anticancer activity	[46]

## 5. Properties of *Ziziphus nummularia*

### 5.1. Anti-Drought and Anti-Thermal Characteristics of Ziziphus nummularia

*Ziziphus* trees and shrubs are able to thrive in extreme habitats, particularly in arid regions (Figure 1) [47]. In the wake of climate change, drought has become a calamitous abiotic stress, affecting plant growth and hampering agricultural production and yield. The economic impact of this stress is likely to increase further; as a result, there has been growing interest in studying drought stress responses and tolerance mechanisms of natural drought-tolerant plants [48], such as *Ziziphus nummularia*. The purpose will be to develop crops with increased tolerance and resilience to drought. In a study to explore the physiological basis of drought, *Ziziphus nummularia* plants were subjected to moisture stress and results showed that chlorophyll *a* and *b* content in leaves decreased, and electrolyte leakage, which normally accompanies the plant’s response to stress, was increased [49]. Furthermore, in another effort to understand the molecular basis of abiotic stress tolerance mechanisms, isolate drought-responsive genes, and eventually develop drought resistant crop varieties, transcriptome profiling and gene expression analysis were carried out on *Ziziphus nummularia* seedlings subjected to drought stress caused by treatment with 30% polyethylene glycol (PEG 6000) [50,51]. In line, Padaria et al. [52] characterized the *Ziziphus nummularia* abscisic acid-stress-ripening gene 1 (*ZnAsr1*), and validated its transcriptional pattern in response to drought stress. This gene was upregulated in response to drought stress, and when recombinantly expressed in *Escherichia coli*, it allowed the bacteria to survive better in PEG-containing media, suggesting that the *ZnASR1* may be used to develop drought tolerant transgenic crops. 

Sivalingam et al. [53] screened the parameters contributing to drought tolerance by comparing three *Ziziphus nummularia* ecotypes from semi-arid, arid and hyper-arid regions. Results showed that the latter was the most drought-tolerant ecotype and was associated with an increase in root length and number, an increase in leaf hair growth, and in elongation of thorns. The leaves were reduced in size and curled, while older leaves fell and shoot growth was sustained. In order to maintain a high relative water content for longer periods of time, membrane permeability was increased, and proline, catalase and sugar content was higher. Transcriptome profiling showed that genes associated with drought response mechanisms were constitutively expressed at higher levels [53]. 

*Ziziphus nummularia* also show tolerance to heat stress. Here, the chaperone protein ClpB1, which plays an important role in thermotolerance, was investigated [54]. It was found that the transcript levels of the *JClpB1*-C isoform from a heat-tolerant *Ziziphus nummularia* ecotype was highly up-regulated under heat stress conditions and that the overexpression of *ZnJCIpB1*-C in tobacco showed a significant increase in its heat tolerance by exhibiting higher photosynthetic rates, relative water content, chlorophyll content, and membrane stability index, suggesting that *Ziziphus nummularia* could provide an important source for developing heat-stress tolerant germplasms. Therefore, *Z. nummularia* can have an industrial potential as it enhances drought resistance in certain areas as well as having heat stress tolerance.

### 5.2. Pharmacological Properties of Ziziphus nummularia

In the past few years, several studies were carried out to investigate and demonstrate the pharmacological and biological properties of various parts of the *Ziziphus nummularia* plant. These are summarized in Figure 2.

#### 5.2.1. Antimicrobial Activity

Antimicrobial resistance is nowadays a major global challenge. It occurs when infectious agents such as bacteria, viruses, fungi, and parasites become resistant to standard treatment, instigating a public health threat by spreading infectious diseases and causing huge global economic losses through food spoilage and damages to crops [55,56]. In order to overcome antimicrobial resistance, researchers, in their search for new drugs, are showing increased interest in medicinal plants due to their rich diversity in secondary metabolites and phytochemicals [57]. The antimicrobial properties of *Ziziphus nummularia* from several studies have been analysed and listed in Table 3. Using the agar well diffusion method, Beg et al. compared the antimicrobial activity of *Ziziphus nummularia* extracts from fruits, leaves and bark against the gram-positive strain *Staphylococcus aureus* and the gram-negative strain *Escherichia coli* using different extraction solvents [58]. Results showed that the methanolic extract, followed by hexane and chloroform extracts, of the fruits had the strongest activity against both of the tested strain; the aqueous extract of all plant parts did not show any antimicrobial activity [58]. Interestingly, it was found that the fruit part of the plant displayed substantial antibacterial activity against various gram-positive strains tested [59]. Another study demonstrated that the chloroform and ethyl acetate fractions of *Ziziphus nummularia* were effective against both gram-negative and gram-positive bacteria, but the methanol and aqueous fractions did not exhibit any activity towards any of the tested microorganisms [41]. Gautam et al. demonstrated that the ethanolic and aqueous *Ziziphus nummularia* extracts exhibited significant activity against *Staphylococcus aureus* and *Pseudomonas aeruginosa*, although it had no activity against *Bacillus subtilis,* noting that the ethanolic extract was more active than the aqueous one [60]. Sharma et al. attributed the antimicrobial activity of *Ziziphus nummularia* to the presence of alkaloids, flavonoids, glycosides, and saponins in the leaf extracts of the plant, proposing that leaf extracts may be an effective agent for the treatment of several diseases [61]. 

Besides, Pandey and Devi showed that the difference in antimicrobial activity was associated with variations in the cyclopeptide alkaloid composition [62]. For instance, the 13-membered cyclopeptide alkaloids, nummularine-R, nummularine-S, and nummularine-B, were effective antibacterial agents only against gram-negative bacteria such as *Klebsiella pneumonia* and *Escherichia coli*, while the 14-membered cyclopeptide alkaloid, Frangufoline, showed significant antibacterial activity against the gram-positive bacterium *Staphylococcus aureus* as well as the gram-negative bacteria *Klebsiella pneumonia* and *Escherichia coli* [62]. 

With regard to the antifungal activity of *Zizuphus nummularia*, it was shown that the ethanolic extract had strong activity against *Aspergillus niger*, *Aspergillus flavus*, *Candida albicans*, and particularly against *Trichophyton rubrum* [60]. All these data show that *Ziziphus nummularia* could be a potential source of novel antimicrobial agents that would support botanical screening efforts in the search for new drugs to overcome antimicrobial resistance and ensure antibiotic-free control of microbial growth in the future.

#### 5.2.2. Anthelmintic Activity

Anthelmintic synthetic drugs are often associated with many side effects, toxic when inappropriately administered, and worrisome regarding the accumulation of drug residues in animals. Moreover, nematodes have developed resistance against many of these drugs, further fuelling interest in plants as sources for anthelmintic bioactive compounds [63]. 

*Ziziphus nummularia* has been traditionally used in veterinary medicine in an effort to control parasitic helminths in small ruminants and livestock. In line, Bachaya et al. have shown that *Ziziphus nummularia* crude methanolic extract is a rich source of chemicals effective against *Haemonchus contortus*, one of the most pathogenic nematodes of ruminants. The extract increased the mortality of worms, and inhibited egg hatching and larval development [64]. Its activity was also tested in vivo, where the same extract exhibited an 84.7% reduction in faecal egg count 13 days after treatment in sheep, a reduction comparable to that obtained with the synthetic drug levamisole [63]. This suggests that *Ziziphus nummularia* provides an economical and safe approach to search for new anthelmintics. It would be interesting to identify the constituents that are active against pathogenic nematodes, study the structure-activity relationship, and explore the underlying molecular mechanisms in order to assess their potential for drug discovery. Table 4 illustrates the anthelmintic activity of *Ziziphus nummularia*.

#### 5.2.3. Antioxidant Activity

Aging and many human diseases, such as cancer, inflammatory disorder, and neurodegenerative and digestive diseases, are associated with the overproduction of reactive oxygen species (ROS) or other free radicals [65,66,67]. These molecules are very reactive because of their unpaired electrons and, as a result, can cause cellular damage. Excessive amounts of ROS are mainly regulated by either endogenous antioxidant cellular mechanisms or antioxidants supplemented from exogenous sources. Hence, there has been an increasing interest in natural exogenous sources of antioxidants including plant sources. Plant antioxidant sources are regarded as a safer alternative to synthetic antioxidants such as butyl hydroxy anisole (BHA) and butylated hydroxytoluene (BHT) [68,69]. In this respect, several studies have been carried out to assess the antioxidant activity of *Ziziphus nummularia*. A summary of the analysis of these properties are listed in Table 5. The methanolic and acetone extracts from the leaves of *Ziziphus nummularia* showed significant scavenging activities of the free radicals 1,1-diphenyl-2-picryl-hydrazyl (DDPH) and superoxide anion radicals. In addition, a high reducing power was recorded, supporting the strong antioxidant potential of this plant [70]. In another study, the methanolic extract from the fruit part of *Ziziphus nummularia*, which contains a high phenolic and flavonoid content, was shown to have a significant antioxidant activity against DDPH and hydrogen peroxide (H_2_O_2_) [71]. Similar results were obtained with the hydro-alcoholic extract from the stem bark, rich in phenolics [72], and with the methanolic extract of roots [42]. Finally, an investigation of the antioxidant capacity of methanolic extracts from different *Ziziphus nummularia* genotypes showed that the antioxidant potential was directly correlated with the extract’s phenolic content [43]. Hence, *Ziziphus nummularia* exhibits potent antioxidant capacities that warrant further investigation. Recently, there has been a growing interest in green extraction techniques that yield increased concentrations of antioxidant compounds from plants in the prevention of oxidative-stress related disorders and promotion of longevity, limiting the effect of aging.

#### 5.2.4. Anti-Inflammatory Effect

Inflammation is part of the body’s homeostatic and self-defence mechanism, and is always kept under homeostatic regulation. However, when inflammation becomes excessive or chronic, it may contribute to the development of several diseases such as diabetes, cardiovascular disease, and cancer [73,74]. Several studies have aimed to explore the reported use of *Ziziphus nummularia* as an anti-inflammatory agent in traditional medicine [16]. An analysis of the anti-inflammatory activities of *Ziziphus nummularia* is listed in Table 6. Using the carrageenan induced paw edema test, the topical application of a gel ethanolic extract from *Ziziphus nummularia* leaves to Wistar albino rats caused a significant reduction in paw edema in a dose-dependent manner, confirming the anti-inflammatory effect of the extract [75]. It also showed faster wound repair abilities when compared to the marketed formulation Betadine^®^ [75]. Dey Ray et al. orally administered the ethanolic extract from the root bark into mice and found it to cause a significant inhibition of carrageenan and arachidonic acid induced edema, and reduce the formation of granuloma tissue, further confirming the anti-inflammatory properties of the plant [76]. Moreover, octadecahydro-picene-2,3,14,15-tetranone, a compound isolated from the root bark of *Ziziphus nummularia*, had a more pronounced anti-inflammatory than the extract, nearly comparable to that of aspirin. It was suggested that this compound controls inflammation through the inhibition of TNF-α and nitric oxide production [76]. Goyal et al. found that the anti-inflammatory effect of the ethanolic extract of *Ziziphus nummularia* was due to the inhibition of histamine release [77], which initially causes vasodilation and increases vascular permeability. This extract also inhibited peritoneal leukocyte migration [77]. Finally, a recent study demonstrated that the ethanolic extract of the leaves of *Ziziphus nummularia* exhibited an anti-inflammatory effect in human aortic smooth muscle cells by decreasing their proliferation, and invasion and migration potentials. Importantly, the extract caused a concentration- and time-dependent decrease in the pro-inflammatory effect of tumor necrosis factor α (TNFα). The extract reduced the TNFα-induced expression of matrix metalloproteases, NF-κB, and cell adhesion molecules [78]. In fact, upon inflammatory stimuli, vascular smooth muscle cells switch from a quiescent phenotype to a synthetic one, entailing detachment of these cells from the extracellular matrix and their migration to the tunica intima layer of the vessel, contributing to the formation and growth of atherosclerotic lesions. By this, the study suggests that *Ziziphus nummularia* extracts reduce inflammation and thereby may reduce inflammation-induced atherogenic atherosclerosis [78].

#### 5.2.5. Anticancer Activity

Cancer continues to be one of the leading causes of mortality worldwide despite great advancements in cancer therapy. Moreover, conventional treatment regimens are often associated with adverse effects and multidrug resistance that have resulted in increasing interest in the search for new bioactive compounds from plant sources [80]. In this regard, several studies have assessed the anticancer properties of *Ziziphus nummularia*. Their analysis is included in Table 7. Lapachol (2-hydroxy3-(3-methy1-2-buteny1)-1,4-naphthoquinone), a naphthoquinone compound isolated from *Ziziphus nummularia*, was shown to have a strong antitumor activity in female Swiss albino mice engrafted with sarcoma ascetic tumor cells (S-180 cells). Lapachol also enhanced the response of the engrafted tumors to radiation therapy, suggesting that it can have a potential use as an adjuvant in radiation therapy [46]. Another phytoconstituent isolated from the root bark of *Ziziphus nummularia* is a new triterpene derivative referred to as Identified Compound (IC) [45]. IC showed high cytotoxicity in several human cancer cell lines in vitro, including breast cancer, leukaemia, ovarian cancer, colon adenocarcinoma, and kidney carcinoma. This anticancer activity was higher compared to the plant’s crude ethanolic extract. In vivo, both IC and the ethanolic extract were administered to mice, engrafted with Ehrlich ascites carcinoma, and were found to decrease tumor parameters including decreasing the viable cancer cell count, restoring serum biochemical parameters, and decreasing ascitic fluid volume, thereby inhibiting cancer growth and increasing the life span of mice. The treatment also decreased the count of white blood cells, increased the count of red blood cells, and elevated haemoglobin content [45]. In another study, the methanolic fruit extract of *Ziziphus nummularia* showed appreciable cytotoxic activity against cervical carcinoma cells, HeLa cells [58]. Furthermore, the ethanolic extract from *Ziziphus nummularia* leaves was shown to inhibit the hallmarks of pancreatic cancer by interfering with tumorigenic and metastatic events [81]. Indeed, treatment of capan-2 pancreatic cancer cells resulted in an attenuation of cell proliferation and a significant decrease in their migration and invasion potentials. The same extract was also able to attenuate angiogenesis by reducing the production of secreted and intracellular vascular endothelial growth factor (VEGF), a pro-angiogenic protein, as well as the levels of nitric oxide. This anti-angiogenic effect was further confirmed by in ovo experiments using the chick embryo chorioallantoic membrane assay [81]. These studies have shown *that Ziziphus nummularia extracts* and the isolated compounds described above exhibit strong anticancer activities against various types cancers and target the hallmarks of cancer. As such, *Ziziphus nummularia* proves to be an effective and safe source of bioactive compounds with anticancer properties. 

#### 5.2.6. Antidiabetic Effect

Diabetes, a very common chronic disease associated with a deregulation of blood sugar, has been on the rise, especially in low- and middle-income countries, and is associated with significant mortality and morbidity rates. Over time, diabetes, if left uncontrolled, can cause serious damage to the blood vessels and nerves, leading to blindness, kidney failure, and cardiovascular complications among others [82]. Before the discovery of insulin, medicinal plants were widely used in the treatment of diabetes; today diabetic patients are showing more interest in herbal remedies and natural compounds due to their cost-effectiveness, availability, and being regarded as safe with limited side effects [83]. As such, the antidiabetic potential of *Ziziphus nummularia* has been tested in several studies, as tabulated in Table 8. As prolonged hyperglycaemia during diabetes leads to abnormalities in lipid profiles and is associated with atherosclerosis, Rajasekaran et al. investigated the anti-hyperglycaemic and hypolipidemic effects of *Ziziphus nummularia* ethanolic and aqueous leaf extracts in alloxan-induced diabetes in rats [84]. Results showed that both extracts, the ethanolic extract in particular, reversed pancreatic damage, significantly decreased blood sugar levels, and reduced the levels of triglycerides (TGL), low density lipoprotein (LDL), very low density lipoprotein (VLDL), and total cholesterol (TC). Levels of cardioprotective high density lipoprotein (HDL) increased [84]. Similar results were obtained in another study [85] and in a dexamethasone-induced diabetic rat model [86], further supporting the importance of *Ziziphus nummularia* as a source of anti-hyperglycaemic and hypolipidemic agents in the treatment of diabetes. Dubey et al. tested the anti-diabetic potential of the aqueous, methanolic, and saponin extracts of the leaves and found that all the extracts, particularly the saponin extract, had a strong anti-diabetic activity that inhibited α-amylase. They proposed that *Ziziphus nummularia* may be a good source for pharmacologically active agents of the metabolism of carbohydrates, and hence the management of diabetes [87]. Furthermore, a comparison of the anti-diabetic activity of hydro-alcoholic extracts from the leaves and fruits of *Ziziphus nummularia* using the Soxhlet and maceration methods was carried out. All extracts showed significant inhibitory activity, but leaf extracts from the maceration process showed the strongest activity and were associated with the highest phenolic content [88]. 

#### 5.2.7. Anticholinesterase Activity 

Alzheimer’s disease is a neurodegenerative disorder of the brain, clinically characterized by a progressive decline in cognitive abilities, and is the most common cause of dementia. It is associated with a significant reduction in central cholinergic neurotransmission due to a loss or deficiency of the neurotransmitter acetylcholine. Restoring acetylcholine levels through the inhibition of the enzymes which hydrolyze acetylcholine, acetyl cholinesterase (AChE) and butyrylcholinesterase (BChE), is used as the main treatment strategy. However, these treatments are often associated with side effects such as nausea and vomiting [89]. As a rich source of phenolic compounds, Uddin et al. investigated the anticholinesterase activity of *Ziziphus nummularia* using methanolic extracts of the fruits of different genotypes [43]. The extracts showed significant activity against both AChE and BChE, suggesting that *Ziziphus nummularia* could be used as a source for potential therapeutic agents in the treatment of neurodegenerative diseases. Differences observed among the different genotypes were attributed to variations in the composition and concentration of their bioactive constituents and secondary metabolites [43]. Table 9 summarizes these activities.

#### 5.2.8. Analgesic and Sedative Activities

*Ziziphus nummularia* has been widely prescribed in traditional medicine for relieving pain. Hence, Goyal et al. investigated its analgesic effects and showed that ethanolic extract from leaves of *Ziziphus nummularia* significantly reduced the number of writhes of mice in a visceral pain model that involved acetic-induced abdominal writhing in mice. The pain relief was attributed to the anti-inflammatory properties of *Ziziphus nummularia* [77,79]. This extract also increased tail flick latency in mice, indicating that its analgesic activity is mediated by central mechanisms [77,79]. Administration of the cyclopeptide alkaloid fraction of the ethanolic extract of the leaves of *Ziziphus nummularia* in mice subjected to acetic acid-induced pain, confirmed the analgesic properties of this plant in addition to possessing significant anti-nociceptive effects [79]. Rauf et al. tested the effects of several fractions of methanolic extracts from roots of *Ziziphus nummularia* in vivo in mice subjected to acetic acid-induced writhing. Results showed that the crude extract remarkably reduced abdominal contractions in a dose-dependent manner, noting that the chloroform fraction was most effective in reducing pain and sensitivity, followed by the ethyl acetate fraction. The authors also suggested a possible mechanism of action that can be attributed to the plant’s anti-inflammatory properties through interactions involving central (opioid and cholinergic) pathways as well as peripheral (COX-2) pain pathways [14]. They also investigated the effect of the sedative-hypnotic effects of the root extract through the open field test, showing that the treated mice had a significant reduction in locomotion and lengthening of sleep durations, specifically when using the chloroform and ethyl acetate fractions [14]. The analgesic and sedative properties of *Ziziphus nummularia* are summarized in Table 10. These results suggest that *Ziziphus nummularia* extracts should be further exploited to isolate compounds with sedative-hypnotic, antipyretic, or analgesic activities. 

#### 5.2.9. Gastrointestinal Properties 

*Ziziphus nummularia* has been traditionally used in the treatment of gut disorders such as diarrhea, gastrointestinal spasms, and ulcer. A study by Sayed et al. was carried out to evaluate these ethnopharmacological properties of the methanolic crude extract of *Ziziphus nummularia* leaves [10]. Results showed that the extract exhibited strong protection against diarrhea and intestinal fluid accumulation in a concentration-dependent manner in mice with castor-oil induced diarrhea; similar to the standard anti-diarrheal drug loperamide [10]. The extract also caused a relaxation of spontaneous KCl-induced contractions in an isolated rabbit jejunum model by blocking Ca^2+^ influx, in a manner similar to that of the drug verapamil [10]. Finally, *Ziziphus nummularia* was shown to also have anti-ulcer properties by decreasing lesions and protecting the gastric mucosa in an ethanol-induced gastric ulceration assay comparable to that of the standard drug omeprazole, which is widely used in the treatment of stomach ulcers as well as indigestion, heart burn and acid reflux [10]. An analysis of the gastrointestinal effects of *Ziziphus nummularia* is presented in Table 11.

#### 5.2.10. Other Activities

Nanotechnology, which involves the development and controlled manipulation of molecules with length scales in the 1- to 100-nanometers range, is an emerging field that is rapidly gaining impetus with applications in drug delivery, molecular imaging, catalysis, and for diagnostic purposes, among others [90]. Recently there has been growing interest in the “green synthesis” of nanoparticles through the use of microbes, plants, or plant parts for the bioreduction of metal ions into their elemental nanoparticle form [91,92,93]. Not only has green synthesis given more stable nanoparticles compared to other methods, but also it has been scalable and eco-friendly without any requirement for high temperatures and pressure or the use of toxic chemicals. So far, *Ziziphus nummularia* leaf extracts have been used in the green synthesis of silver [6,94,95,96], gold [95], and zinc nanoparticles [97] (Figure 3). Indeed, during the formation of nanoparticles, the extract of *Z. nummalaria* functions as a reducing and stabilizing agent [96]. This increases the potential activity of the nanoparticles. 

Silver nanoparticles are mostly used as anti-microbial agents in the medical field mainly due to their strong growth inhibitory effects against fungi, bacteria and viruses, while having low toxicity to human cells. These nanoparticles have been utilized in food storage, textile coatings, as well as water treatment [99]. Khan et al. were the first to synthesize silver nanoparticles using *Ziziphus nummularia* aqueous leaf extract as the reducing and capping agent [94]. Results showed that the prepared nanoparticles exhibited better antioxidant activity compared to the leaf extract. Antibacterial activity against *Escherichia coli*, *Pseudomonas aeruginosa*, *Staphylococcus aureus*, *Salmonella typhi*, and *Bacillus cereus*, as well as antifungal activity against *Aspergillus niger*, *Aspergillus flavus* and *Candida albicans* were also enhanced. Similar enhancements were reported in another study, also using *Ziziphus nummularia* aqueous leaf extracts, where increased antioxidant and antibacterial activity against the cariogenic *Staphylococcus aureus* and *Streptococcus mutans* [95], *Bacillus subtitles*, *Pseudomonas aerugionsa*, *Klebsiella pneumonia*, *Escherichia coli* and *Streptococcus aureus* were reported [6]. These nanoparticles also showed good fungicidal activity against *Aspergillus niger* and *Aspergillus flavus* [6].

Gold nanoparticles, which are also known to possess antimicrobial activities [100], were also synthesized using *Ziziphus nummularia* aqueous leaf extracts. These showed enhanced antioxidant potential and anti-microbial activities, similar to the results obtained with silver nanoparticles [95]. Zinc oxide (ZnO) nanoparticles are widely used in various medical applications such as drug delivery and bio-imaging [101] and are known to exhibit anti-microbial activities [102]. 

The anti-microbial properties of ZnO nanoparticles synthesized with the aid of *Ziziphus nummularia* leaf extracts were tested against clinical isolates of *Candida* species (*Candida albicans*, *Candida glabrata*) and *Cryptococcus neoformans* [97]. Results showed strong fungicidal activity, better than the leaf extract and the standard antibiotic amphotericin B. The prepared ZnO nanoparticles also exhibited a dose-dependent cytotoxic effect against HeLa cancer cells, higher than the *Ziziphus nummularia* leaf extract [97]. 

## 6. Toxicology Studies

Although natural compounds from plants sources have successfully led to the discovery of important drugs that are still currently being used to treat various diseases, many herbal products are poorly tested or monitored, if tested at all, for their efficacy and safety. In fact, herbs as medicinal drugs are not as safe as claimed and certain medicinal herbs could exhibit adverse effects. Consequently, it is important that toxicology studies are carried out and adequate information is provided to consumers on the risks associated with the application of these products [103,104,105]. *Ziziphus nummularia* was not found to exhibit any behavioral changes or mortality in mice administered with aqueous-methanol *Ziziphus nummularia* leaf extract at oral doses of 3 and 5 g/kg, and up to 10 g/kg in a 24-h study, suggesting that *Ziziphus nummularia* is safe to use, and making it an attractive plant for natural drug discovery approaches [10]. In addition, toxicity analysis of other *Ziziphus* species concluded that they were safe [106,107]. Nonetheless, although toxicity studies indicate that *Ziziphus nummularia* extracts are nontoxic, there is still a need for further toxicology screening and particularly for long-term chronic toxicity studies to validate their safe usage.

## 7. Conclusions

Today more than ever, the use of natural compounds from plant sources as potential drug leads is gaining more interest, both in research and the pharmaceutical industry. Phytochemical and pharmacological studies on *Ziziphus nummularia* have shown so far that it can be a rich source of new compounds with wide applications in the pharmaceutical and food industries (Figure 3). For instance, its antimicrobial, antifungal, and anthelminthic properties suggest its potential use as a natural food preservative. Moreover, its antioxidant and antidiabetic properties make it a potential source of agents with antiobesogenic and metabolic benefits. Finally, the various pharmacological properties (anticancer, anti-inflammatory, gastrointestinal, neuroprotective, and cardioprotective) of *Ziziphus nummularia* warrant that the scientific research community and pharmaceutical industries may target this plant as a potential source for the discovery and use of new bioactive molecules. Further research on the mechanism of action, pharmacodynamic, and pharmacokinetic profiles of *Z. nummularia* extract as a medicinal plant should be investigated further, in conjunction with toxicological studies, for the development of natural and safe pharmaceutical drugs.

## Figures and Tables

**Figure 1 molecules-27-04240-f001:**
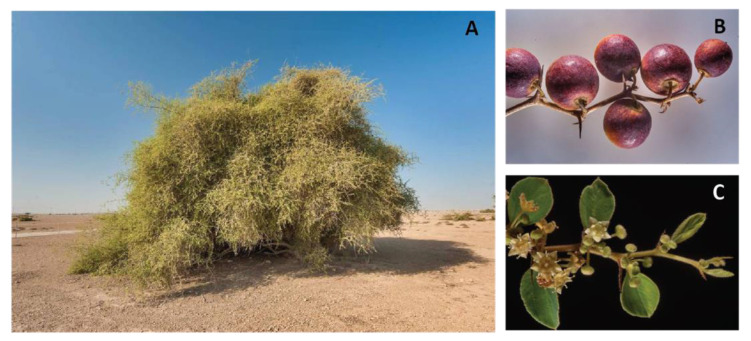
(**A**) *Ziziphus nummularia* plant, (**B**) *Ziziphus nummularia* fruit, and (**C**) *Ziziphus nummularia* leaves. Images were obtained from https://www.floraofqatar.com/ziziphus_nummularia.htm (accessed on 20 June 2022).

**Figure 2 molecules-27-04240-f002:**
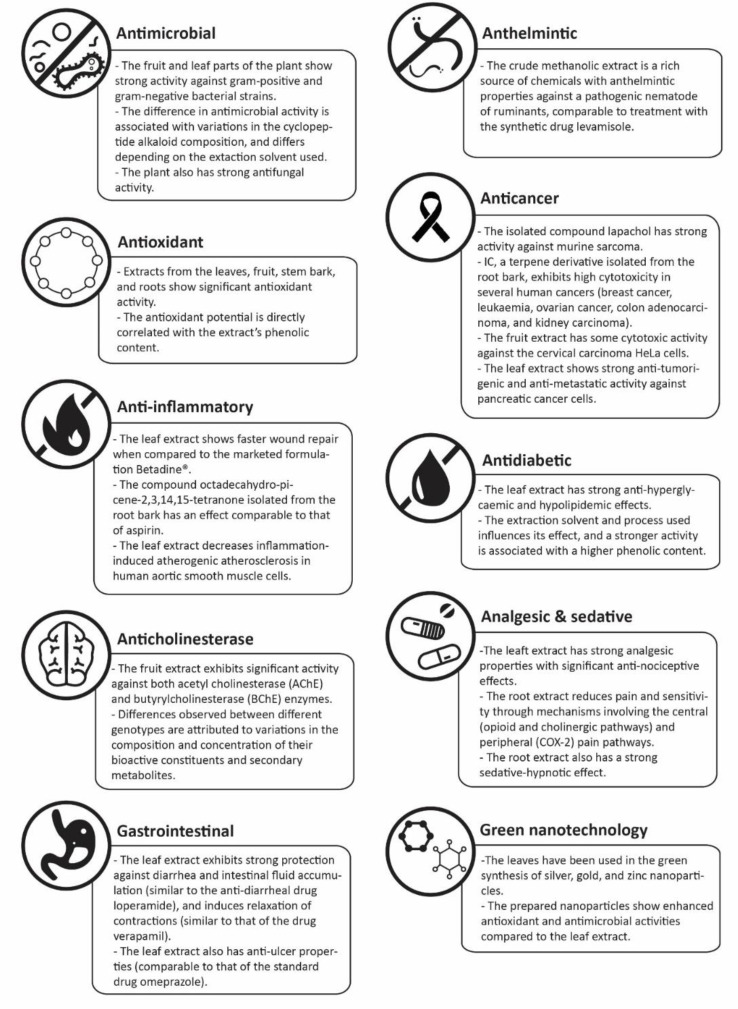
Reported therapeutic applications of *Ziziphus nummularia*.

**Figure 3 molecules-27-04240-f003:**
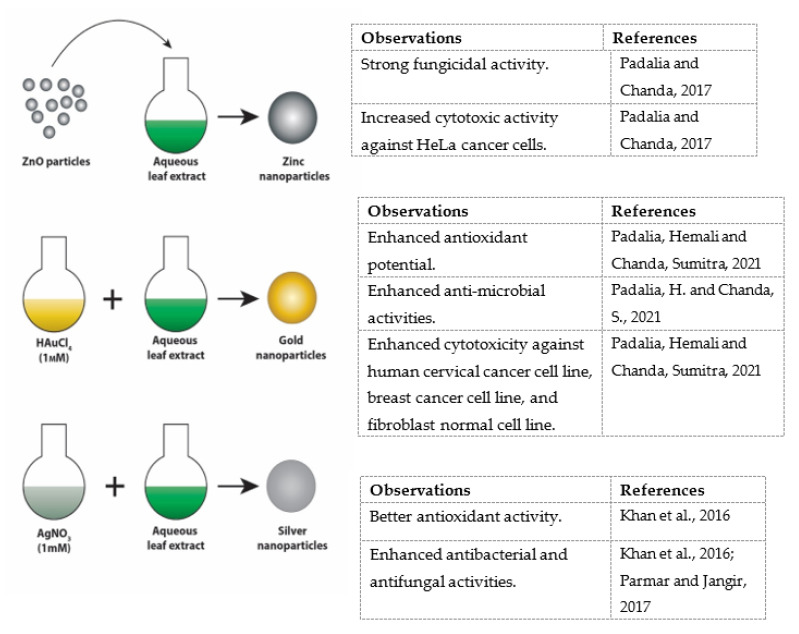
Reported therapeutic applications of *Ziziphus nummularia*. Padalia and Chanda, 2017 [97]; Padalia, H. and Chanda, S., 2021 [98]; Padalia, H. and Chanda, S., 2021 [96], Khan et al., 2016 [94]; Parmar and Jangir, 2017 [95].

**Table 3 molecules-27-04240-t003:** The antimicrobial effects of *Zizphus nummularia*.

Extract	Dose	Experimental Model	Observations	References
** *Antibacterial* **				
**Aqueous and ethanolic extracts**	0.4 parts per million (ppm), 0.8 ppm, 1.6 ppm, and 3.2 ppm)	Strains: *Staphylococcus aureus*, *Streptococcus pyogenes*, *Bacillus subtilis*, and *Pseudomonas aeruginosa* Methods: Agar disk diffusion method and dry weight method	Showed potential antibacterial activity. Most susceptible was *Staphylococcus aureus* Ethanolic extract was more active than the aqueous one.	[60]
**Acetone, benzene, chloroform, petroleum ether, and water extracts from the leaves**	Acetone: 3.7 mg Benzene: 4 mg Chlorofom: 3.45 Petroleum ether: 2.5 Water: 4.8	Strains: *Staphylococcus aureus* and *Escherichia coli* Method: Agar disk diffusion method	No antibacterial activity against *Escherichia coli*. All extracts showed activity against *Staphylococcus aureus*.	[61]
**Crude methanolic from the fruits and fractions**		Strains: 13 g-positive bacterial strains and 15 g-negative bacterial strains Method: Agar disk diffusion method	Higher activity observed against gram-positive strains, with no activity against gram-negative strains with certain fractions	[59]
**Methanolic, aqueous, chloroform and hexane extracts from the fruits, leaves and bark**	100 mg/mL	Strains: *Staphylococcus aureus* and *Escherichia coli* Method: Agar disk diffusion method	The methanolic and hexane extracts from the fruit showed significant antibacterial activity. The chloroform extract from the fruit showed moderate activity. The aqueous extract from all plant parts showed no activity.	[58]
**Methanolic extract from the leaves and fractions**	1 and 2 mg/mL	Strains: *Staphylococcus aureus*, *Bacillus subtillus*, *Listeria monocytogenes*, *Klebsiella pneumoniae*, *Escherichia coli*, and *Pseudomonas aureginosa* Method: Agar disk diffusion method	The chloroform and ethyl acetate fractions showed strongest effect against both gram-negative and positive bacteria. The methanolic and aqueous fraction did show any activity against tested microorganism	[41]
**Methanolic extract from the leaves and fractions**	1 and 2 mg/mL	Strain: *Pseudomonas aureginosa* Methods: Biofilm activity assay	Dose-dependent inhibition observed with all fractions.The n-hexane fraction showed the highest inhibition (88%), followed by ethyl acetate (69%), and chloroform (65%) fraction at concentration of 2 mg/mL	[41]
**Methanolic crude extract from the leaves and fractions**		Strains: *Staphylococcus aureus, Bacillus subtilis* and *Klebsiella pneumonia*Method: Agar disk diffusion method	All the tested solvent fractions showed moderate activity	[42]
** *Antifungal* **				
**Aqueous and ethanolic extracts**	0.4 parts per million (ppm), 0.8 ppm, 1.6 ppm, and 3.2 ppm)	Strains: *Aspergillus niger, Aspergillus flavus, Candida albicans, and Trichophyton rubrum* Method: Agar well diffusion method and dry weight method	Showed potential antifungal activity. Most susceptible was *Trichophyton rubrum*.	[60]
**Methanolic crude extract from the leaves and fractions**		Strains: *Aspergills flavus*, *Aspergills niger* and *Alternaria solani* Method: Tube dilution method	The extract and all fractions showed strong anti-fungal activity, with the n-hexane fraction showing maximum activity.	[42]
**Methanolic extract from the leaves and fractions**	10 mg/mL	Strains: *Aspergills flavus*, *Aspergills niger* and *Alternaria solani* Method: Antifungal assay	None of the fractions, including crude extract inhibited fungal growth	[41]

**Table 4 molecules-27-04240-t004:** The anthelmintic effects of *Ziziphus nummularia*.

Extract	Dose	Experimental Model	Observations	References
**Crude methanolic extract (CME) from the bark**	500–8000 μg/mL of CME	Adult motility assay on mature live *Haemonchus contortus* of sheep	Caused mortality of worms in dose and time-dependent manner	[64]
**Crude methanolic extract (CME) from the bark**	62.5–4000 μg/mL	Egg hatch test	Inhibited egg hatching in dose and time-dependent manner (LC_50_ = 676.08 μg/mL).	[64]
**Crude methanolic extract (CME) from the bark**	62.5–4000 μg/mL	Larval development assay	Inhibited larval development in dose- and time-dependent manner (LC_50_ = 398.11 μg/mL).	[64]
**Crude powder and crude methanolic extract (CME) from the bark**	1.0–3.0 g/kg	In vivo study: sheep naturally infected with gastrointestinal nematodes	Maximum reduction in fecal egg count reduction (84.7%) recorded on day 13 post-treatment in sheep with CME at 3.0 g/kg	[64]

**Table 5 molecules-27-04240-t005:** The antioxidant effects of *Ziziphus nummularia*.

Extract	Dose	Experimental Model	Observations	References
**Acetone and methanolic extracts**		DPPH free radical scavenging assay	Showed strong activity.	[70]
**Acetone and methanolic extracts**		Hydroxyl radical scavenging assay	Showed strong activity.	[70]
**Acetone and methanolic extracts**		Superoxide anion radical scavenging assay	Showed strong activity.	[70]
**Acetone and methanolic extracts**	20–180 μg/mL	Reducing capacity assessment	Exhibited increased reducing power in dose-dependent manner	[70]
**Methanolic extract from the fruits**	50–250 μg/mL	DPPH free radical scavenging assay	Showed significant free radical scavenging activity in a dose-dependent manner (77.5% at 250 μg/mL).	[71]
**Methanolic extract from the fruits**	50–250 μg/mL	H_2_O_2_ free radical scavenging assay	Showed significant free radical scavenging activity in a dose-dependent manner (71% at 250 μg/mL).	[71]
**Methanolic extract from the fruits**	31.25, 62.5, 125, 250, 500, 1000 μg/mL	DPPH free radical scavenging assay	All genotypes showed potent scavenging activity against DPPH. Highest % DPPH inhibition value was 67.03 ± 1.04 mg/mL at 31.25 μg/mL.	[43]
**Methanolic extract from the fruits**	31.25, 62.5, 125, 250, 500, 1000 μg/mL	ABTS Radical Scavenging assay	All genotypes showed potent scavenging activity against ABTS. Highest % ABTS inhibition value was 65.31 ± 1.74 mg/mL at 31.25 μg/mL.	[43]
**Methanolic crude extract from the leaves and fractions (n-hexane, chloroform, ethyl acetate, ethanol)**	10–100 μg/mL	DPPH free radical scavenging assay	The ethyl acetate fraction exhibited maximum (97.77%) free radical scavenging activity at the 100 μg/mL.	[42]

**Table 6 molecules-27-04240-t006:** The anti-inflammatory effects of *Ziziphus nummularia*.

Extract	Dose	Experimental Model	Observations	References
**Ethanolic extract (EE) from the leaves**	0.5 g of formulated gels containing 20% and 30% extract	Carrageenan-induced paw edema in Wister albino rats	Significantly reduced paw edema in dose-dependent manner.	[75]
**EE from the leaves**	0.5 g of formulated gels containing 20% and 30% extract	Excision wound model in wister albino rats	Showed an acceleration in the wound healing process and high wound contraction rate.	[75]
**EE from the leaves**	100, 200 and 300 mg/kg	Carrageenan-induced paw edema in rats	Dose-dependent increase in percentage inhibition of paw edema, with 38.37% inhibition at 300 mg/kg.	[79]
**EE from the leaves**	100, 200 and 300 mg/kg	Histamine-induced inflammation in rats	Dose-dependent increase in percentage inhibition of paw edema, with 42.1% inhibition at 300 mg/kg.	[79]
**EE from the root bark** **and** **isolated identified compound (IC)**	EE: 100 and 200 mg/kg IC: 400 and 600 μg/kg	Carrageenan-induced paw edema in mice	Exhibited significant inhibition of paw edema, with strong activity observed with IC.	[76]
**EE from the root bark** **and** **isolated IC**	EE: 100 and 200 mg/kg IC: 400 and 600 μg/kg	Arachidonic acid-induced ear edema in mice	Exhibited significant inhibition of ear edema, with stronger activity observed with IC.	[76]
**EE from the root bark** **and** **isolated IC**	EE: 100 and 200 mg/kg IC: 400 and 600 μg/kg	Xylene-induced ear edema in mice	Exhibited significant inhibition of ear edema, with stronger activity observed with IC.	[76]
**EE from the root bark** **and** **isolated IC**	EE: 100 and 200 mg/kg IC: 400 and 600 μg/kg	Chronic cotton pellet granuloma models in mice	Showed significant inhibition in granuloma tissue formation, with significant inhibition observed with IC.	[76]
**IC from the root bark**	1–50 μM	Lipopolysaccharide-stimulated RAW 264.7 cells	Increased production of the pro-inflammatory cytokines and inflammatory mediators: NO, PGE-2 and TNF-α.	[76]
**EE from the leaves**	50–200 μg/mL	Human aortic smooth muscle cells (HASMCs)	Decreased FBS-induced HASMC proliferation, migration, invasion, and adhesion to fibronectin. Reduced TNF-α-induced expression of matrix metalloproteases (MMP-2 and MMP-9), NF-κB, and cell adhesion molecules (ICAM-1 and VCAM-1) in dose-dependent manner Decreased the adhesion of THP-1 monocytes in dose-dependent manner.	[78]

**Table 7 molecules-27-04240-t007:** The anticancer effects of *Ziziphus nummularia*.

Extract	Dose	Experimental Model	Observations	References
**Lapachol**		Swiss albino mice with sarcoma-180 (S-180) ascetic tumor cell	Showed strong antitumor activity. Enhanced the response of the engrafted tumors to radiation therapy.	[37]
**Identified compound (IC)**	1–1000 μM	Human breast cancer, leukaemia, ovarian cancer, colon adenocarcinoma and human kidney carcinoma cell lines	Showed high cytotoxicity against all cell lines, with IC having stronger activity compared to EE.	[45]
**Ethanolic extract (EE) from the root bark** **and** **isolated identified compound (IC)**	EE:100 and 200 mg/kg IC: 50 and 100 mg/kg	Female Swiss albino mice with *Ehrlich ascites carcinoma*	Decreased tumor parameters: tumor volume, viable tumor cell count and increased body weight, haematological parameters and life span. Decreased the viable cancer cell count and increased the total cell count of the cancer-bearing mice. Significantly elevated the levels of serum biochemical parameters.	[45]
**Methanolic extract from the fruit**	50–300 μg/mL	HeLa cells (cervical carcinoma cells)	Showed increased cytotoxicity and altered morphology of cancer cells.	[58]
**EE from the leaves**	100–300 μg/mL	Human pancreatic cancer	Inhibited cell proliferation, migration, invasion, adhesion, and angiogenesis, and it increased cell-cell aggregation. Inhibited ERK1/2(MAPK) and NFκB signaling pathways.	[81]

**Table 8 molecules-27-04240-t008:** The antidiabetic effects of *Ziziphus nummularia*.

Extract	Dose	Experimental Model	Observations	References
**Ethanolic and aqueous extracts from the leaves**	250 and 500 mg/kg	Dexamethasone induced diabetic rat model	Maintenance of body weight throughout the experiment in treated group. Showed significant decrease in blood glucose level, particularly with the ethanolic extract at 500 mg/kg. Showed significant reduction in diabetes-induced hyperlipidemia.	[84,86]
**Ethanolic extract from the leaves**	250 and 500 mg/kg	Alloxan-induced rat diabetic model	Maintenance of body weight throughout the experiment in treated group. Showed significant reduction in blood glucose levels, observed on day 3 and day 7 with the 500 mg/kg and 250 mg/kg doses, respectively.	[85]
**Ethanolic extract from the leaves and fruits**	Not determined	Glucose diffusion inhibitory test out of a hen’s egg dialysis membrane	Leaf and fruit extract showed significant inhibitory activity, maximum inhibition observed with with leaf extract of the soxhlet process.	[88]
**Aqueous, methanolic and saponin extracts from the leaves**	80–160 μg/mL	α-amylase inhibition assay	All extracts showed strong inhibitory activity with maximum inhibition observed with the saponin extract.	[87]

**Table 9 molecules-27-04240-t009:** The anticholinesterase effects of *Ziziphus nummularia*.

Extract	Dose	Experimental Model	Observations	References
**Methanolic from the fruits**	31.25, 62.5, 125, 250, 500, 1000 μg/mL	In-practice method for assessing the ability to inhibit AChE and BChE	All tested genotypes displayed significant inhibitory effects (highest AChE and BChE inhibition observed with IC_50_ = 20.52 μg/mL 22.76 μg/mL, respectively)	[43]

**Table 10 molecules-27-04240-t010:** The analgesic and sedative effects of *Ziziphus nummularia*.

Extract	Dose	Experimental Model	Observations	References
Ethanolic from the leaves	100, 200 and 300 mg/kg	Mouse carrageenan peritonitis in mice	Dose-dependent inhibition of peritoneal leukocyte migration.	[79]
Ethanolic from the leaves	100, 200 and 300 mg/kg	Acetic acid-induced writhing response in mice	Dose-dependent decrease in the number of writhes, with 59.29% inhibition at 300 mg/kg.	[79]
Ethanolic from the leaves	100, 200 and 300 mg/kg	Tail-flick reaction in mice	Significantly increased tail flick latency, with 90% antinocieptive activity observed at 300 mg/kg.	[79]
Crude methanolic from the roots and fractions	50 and 100 mg/kg	Sedative activity using the open field method in mice	Marked sedative effect with decreased movement in dose-dependent manner. Strongest effect observed with the chloroform fraction.	[14]
Crude methanolic from the roots and fractions	50 and 100 mg/kg	Phenobarbitone-induced sleeping time in mice	Dose-dependent significant reduction in the sleep latency time (time taken for the onset of sleep) and increase in the sleep. Strongest effect observed with the chloroform fraction.	[14]
Crude methanolic from the roots and fractions	50 and 100 mg/kg	Brewer’s-yeast-induced hyperthermia	Pronounced reduction in induced pyrexia. Strongest effect observed with the chloroform fraction.	[14]
Crude methanolic from the roots and fractions	50 and 100 mg/kg	Acetic acid-induced writhing test in mice	Significant reduction pain sensation. Strongest effect observed with the chloroform fraction.	[14]

**Table 11 molecules-27-04240-t011:** The gastrointestinal effects of *Ziziphus nummularia*.

Extract	Dose	Experimental Model	Observations	References
Crude extract from the leaves	50, 100 and 300 mg/kg	Castor oil-induced diarrhea in mice	Exhibited a protective effect against castor oil-induced diarrhea.	[10]
Crude extract from the leaves	300 and 1000 mg/kg	Enteropooling assay for intestinal fluid accumulation in mice	Showed a protective effect on intestinal fluid accumulation.	[10]
Crude extract from the leaves	0.01–3 mg/mL	KCl (80 mM)-induced contractions in isolated rabbit jejunum tissues	Caused a concentration-dependent relaxation of spontaneous and KCl-induced contractions.	[10]
Crude extract from the leaves	300 and 1000 mg/kg	Ethanol-induced gastrointestinal ulcer model	Caused 52.5 and 93.6% inhibition of gastric lesions, respectively.	[10]

## Data Availability

Not applicable.

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
