# Peer review of "Ziziphus nummularia*: A Comprehensive Review of Its Phytochemical Constituents and Pharmacological Properties"

_molecules, 2022, doi:10.3390/molecules27134240_

Round 1
Reviewer 1 Report
This article focuses on the botany, phytochemistry, pharmacological activity and toxicity of Ziziphus nummularia. From the perspective of topic selection, Ziziphus nummularia has reported its national application and significant biological activity. The author summarized and introduced the relevant research of ziziziphus nummularia, which is of great significance to the development of Ziziphus nummularia. The structure of the article is clear, the content is familiar, and the language is good. However, there are still some shortcomings. According to my point of view, some suggestions are given as follows:
The first point is about references. The format of references in this paper is not very standard, for example: 26 Gupta, D.; Mann, S.; Jain, I.; Gupta, R. PHYTOCHEMICAL, *UTRITIO*AL A*D A*TIOXIDA*T ACTIVITY EVALUATIO* 566 OF FRUITS OF ZIZIPHUSUMMULARIA BURM. F. 2011., Please also modify according to the requirements of the magazine.
Second, Ziziphus nummularia should use italics in the summary.
The third point is about the chemical part. The chemical structure needs to be beautified and standardized. We obviously feel that the bond thickness of alkaloid structures such as nummularine N and nummularine P is not consistent. It is suggested to recheck all structures.
The fourth point is table 4 the antagonistic effects of Ziziphus nummularia. The header font is inconsistent with table 1, table 2 and table 3. Table 9 has the same problem.
In general, the content of this article is clear and rich. I think it can be published after some format problems are modified.
Author Response
Dear Reviewer,
We would like to thank you for your positive comments and suggestions that allowed us to improve our manuscript.
Please find below our answers.
Regards
Dr Marc Maresca
Reviewer 1:
This article focuses on the botany, phytochemistry, pharmacological activity and toxicity of Ziziphus nummularia. From the perspective of topic selection, Ziziphus nummularia has reported its national application and significant biological activity. The author summarized and introduced the relevant research of ziziziphus nummularia, which is of great significance to the development of Ziziphus nummularia. The structure of the article is clear, the content is familiar, and the language is good. However, there are still some shortcomings. According to my point of view, some suggestions are given as follows:
- The first point is about references. The format of references in this paper is not very standard, for example: 26 Gupta, D.; Mann, S.; Jain, I.; Gupta, R. PHYTOCHEMICAL, *UTRITIO*AL A*D A*TIOXIDA*T ACTIVITY EVALUATIO* 566 OF FRUITS OF ZIZIPHUSUMMULARIA BURM. F. 2011. Please also modify according to the requirements of the magazine.
We thank the Reviewer for noting this. All references were checked for completeness and adjusted where necessary. In addition, the MDPI citation style was applied.
- Second, Ziziphus nummularia should use italics in the summary.
All Z. nummalria are now italicized in the new manuscript.
- The third point is about the chemical part. The chemical structure needs to be beautified and standardized. We obviously feel that the bond thickness of alkaloid structures such as nummularine N and nummularine P is not consistent. It is suggested to recheck all structures.
We thank the reviewer for this comment. All structures have been checked and modified in the new manuscript as suggested.
- The fourth point is table 4 the antagonistic effects of Ziziphus nummularia. The header font is inconsistent with table 1, table 2 and table 3. Table 9 has the same problem.
Header fonts of Tables 4 and 9 are now made consistent with header of rest of tables. Thank you.
- In general, the content of this article is clear and rich. I think it can be published after some format problems are modified.
We thank the Reviewer for his effort, time, comments, and suggestions which will enhance that quality of the manuscript.
Reviewer 2 Report
The review article by J. Mesmar et al. describes the chemical and pharmacological properties of the plant Ziziphus nummularia covering the literature of the past 20 years.
The review is comprehensive and contains interesting activities of the plant extracts, such as drought resistance, and therapeutical potential due to antioxidant, antiflammatory, anticancer, antimicrobial, antiparasite and other activities. It is also included the use of the plants extracts in the green synthesis of nanoparticles with enhanced properties.
I recommend the publication of this review article in Molecules after a minor revision.
I have the following comments for the authors:
1 1) It is not clear in the text whether previous reviews exist in the literature concerning the same plant (e.g. reference 17). A statement about the existence of previous reviews should be included.
2 2) The chemical structures in Table 2 should be generated by the same program keeping the same settings i.e. bond lengths, atom labels, etc. The structures of molecules belonging to the same family are expected to be written in a consistent way indicating the stereochemistry of the chiral centers.
3 3) Correction of the following phrases:
· Page 1, line 47: to having an anti-infective and anti-cancerous potential
Correct to: having an anti-infective and anti-cancer potential
· Page 2, line 50: benzenoids
Rather to replace with aromatic or polyaromatic compounds
· Page 2, line 62: although these have not been backed by scientific proof yet
Correct to: backed up or supported
· Page 3, line 112: The word „from“ should be removed
· Page 12, line 196: ethanolic extract was more active than aqueous one
Correct to: than the aqueous one
· Page 15, line 245: These molecules very reactive
Correct to: these molecules are very reactive
· Page 15, line 263: there has been growing interest
Correct to: there has been a growing interest
4) Several references are incomplete.
Author Response
Dear Editor, Dear Reviewer,
We would like to thank you for your positive comments and suggestions that allowed us to improve our manuscript.
Please find below our answers.
Regards
Dr Marc Maresca
Reviewer 2:
The review article by J. Mesmar et al. describes the chemical and pharmacological properties of the plant Ziziphus nummularia covering the literature of the past 20 years. The review is comprehensive and contains interesting activities of the plant extracts, such as drought resistance, and therapeutical potential due to antioxidant, antiflammatory, anticancer, antimicrobial, antiparasite and other activities. It is also included the use of the plants extracts in the green synthesis of nanoparticles with enhanced properties. I recommend the publication of this review article in Molecules after a minor revision. I have the following comments for the authors:
- It is not clear in the text whether previous reviews exist in the literature concerning the same plant (e.g. reference 17). A statement about the existence of previous reviews should be included.
The Reviewer is right. There have been several reviews of this topic. But, this manuscript presents a comprehensive review of the topic by including most of the reported activities of Z. nummalria and also gives precious information about the molecules responsibles for such activities, at which doses and by which mechanisms when known. As per recommendation of the Reviewer, we have added that this topic was reviewed by others and we’ve listed 3-4 important and recent reviews references in lines 75-77.
- The chemical structures in Table 2 should be generated by the same program keeping the same settings i.e. bond lengths, atom labels, etc. The structures of molecules belonging to the same family are expected to be written in a consistent way indicating the stereochemistry of the chiral centers.
We thank the reviewer for this suggestion. Table 2 has been modified in the new manuscript as suggested.
- Correction of the following phrases:
- Page 1, line 47: to having an anti-infective and anti-cancerous potential. Correct to: having an anti-infective and anti-cancer potential
- Page 2, line 50: benzenoids. Rather to replace with aromatic or polyaromatic compounds
- Page 2, line 62: although these have not been backed by scientific proof yet. Correct to: backed up or supported
- Page 3, line 112: The word „from“ should be removed
- Page 12, line 196: ethanolic extract was more active than aqueous one. Correct to: than the aqueous one
- Page 15, line 245: These molecules very reactive. Correct to: these molecules are very reactive
- Page 15, line 263: there has been growing interest. Correct to: there has been a growing interest
We thank the reviewer for pointing these mistakes out. All phrases have been corrected as suggested.
- Several references are incomplete
We thank the Reviewer for noting this. All references were checked for completeness and adjusted where necessary.
Reviewer 3 Report
This review article discusses Ziziphus nummularia: A comprehensive review of its phytochemical constituents and pharmacological properties. Before recommending this article for publication, there are some shortcomings for that should be resolve.
General comments
Use accepted complete name of the species.
Species names must be italicized in whole MS.
Abstract
This section is well written but conclusion should be containing one sentence on future recommendation.
Introduction
Introduction is well written; however, some information could be added to further improve.
Add a general paragraph economic, medicinal and historical perspective of plant uses.
As the study title include photochemistry. So add some information on chemicals or drugs derived from the studied species.
Line 135 “mechanisms of natural drought-tolerant plants” the articles reference could be cited https://doi.org/10.3390/ijms23084450
Add mechanism of anti-oxidant and anti-inflammatory responses.
Line 430-436 the following articles could be added
https://doi.org/10.1002/jemt.23553,
Add mechanism of plants influencing NPs synthesis.
Add future perspective of this study. Also add potential of the studies species as industrial important plant and facilitating drought resistance in certain areas.
This manuscript can be accepted after recommended revision.
Author Response
Dear Editor, Dear Reviewer,
We would like to thank you for your positive comments and suggestions that allowed us to improve our manuscript.
Please find below our answers.
Regards
Dr Marc Maresca
Reviewer 3:
This review article discusses Ziziphus nummularia: A comprehensive review of its phytochemical constituents and pharmacological properties. Before recommending this article for publication, there are some shortcomings for that should be resolve.
General comments
- Use accepted complete name of the species.
The Scientific classification of all species used in this manuscript were adopted from according to the nomenclature of “Index Kewensis”. This is discussed in the manuscript lines 87-90 pages 2.
- Species names must be italicized in whole MS.
All species name are now italicized in the new manuscript.
Abstract
- This section is well written but conclusion should be containing one sentence on future recommendation.
Future recommendation statement has been added at the end of the Abstract. Thank you.
Introduction
- Introduction is well written; however, some information could be added to further improve. Add a general paragraph economic, medicinal and historical perspective of plant uses.
A paragraph was added to the manuscript on Pages 1-2 lines 42-50 as suggested by the reviewer.
- As the study title include photochemistry. So add some information on chemicals or drugs derived from the studied species.
We thank the reviewer for this recommendation. We have already discussed some of the phytochemicals present in Z. nummalaria. For example, nummularine-N, page 3 line 106, Nummularine P page 3 line 108, and a summary of the phytochemicals derived from Z. nummalaria is present in tables 1 and 2.
- Line 135 “mechanisms of natural drought-tolerant plants” the articles reference could be cited https://doi.org/10.3390/ijms23084450
This reference has been added as ref#54
Add mechanism of anti-oxidant and anti-inflammatory responses.
These mechanisms are mentioned in the manuscript on Page 15 lines 266-267 and Page 17 lines: 298-299, 301 and 306.
Line 430-436 the following articles could be added: https://doi.org/10.1002/jemt.23553
This reference has been added as ref#109.
- Add mechanism of plants influencing NPs synthesis.
Add future perspective of this study.
Also add potential of the studies species as industrial important plant and facilitating drought resistance in certain areas.
These mechanisms are now added to the manuscript on Page 24 lines 453-454 as suggested by the reviewer.
Future recommendations/ applications are also added Page 25 511-513
The industrial importance of this plant has been added on page 10 lines 179-180.
This manuscript can be accepted after recommended revision.
Round 2
Reviewer 3 Report
No further comments